# Functional Ability, Frailty and Risk of Falls in the Elderly: Relations with Autonomy in Daily Living

**DOI:** 10.3390/ijerph17031006

**Published:** 2020-02-05

**Authors:** Inmaculada Tornero-Quiñones, Jesús Sáez-Padilla, Alejandro Espina Díaz, Manuel Tomás Abad Robles, Ángela Sierra Robles

**Affiliations:** Faculty of Education, Psychology and Sport Sciences, University of Huelva, 21071 Huelva, Spain; inmaculada.tornero@dempc.uhu.es (I.T.-Q.); alejandro.espina@ddi.uhu.es (A.E.D.); manuel.abad@dempc.uhu.es (M.T.A.R.); sierras@uhu.es (Á.S.R.)

**Keywords:** older people, quality of life, exercise, prevention, falls

## Abstract

The objective of this research was to determine the differences in autonomy in both basic activities of daily life in instrumental activities of daily life, as well as functional capacity, fragility and risk of falls between an active group and a sedentary group. The individual associations of functional capacity, fragility and risk of falls were also analyzed, with autonomy in basic activities of daily living and in instrumental activities of daily living in the active group. In this cross-sectional investigation, 139 people from Huelva between 65 and 87 years of age were evaluated (Mean (M) = 73.1; standard deviation (SD) = 5.86); 100 were women and 39 men. The active and sedentary group were composed of 69 and 70 elderly people, respectively. The active group carried out a physical activity program. Among the results, a significant effect was seen in the multivariate contrast of the study variables, *V* = 0.24, *F* (5, 137) = 8.58, and *p* < 0.001; while in the linear regressions in the active group, the Vivifrail with the Barthel Index (Δ Adj. R^2^ = 0.15) and with the Lawton and Brody Scale (Δ Adj. R^2^ = 0.22) were used. In conclusion, the active group presented better values in all the variables evaluated in comparison to the sedentary group, establishing statistically significant differences. In addition, in the active group, it has been found that functional capacity is a significant predictive variable of autonomy in instrumental activities of daily living (22%), while fragility and the risk of falls are significant predictors of autonomy in activities of basic daily life (15%).

## 1. Introduction

Currently, the ageing demographic plays a large role in developed societies [1] and are becoming a highly relevant social and political challenge [2]. This demographic change has occurred due to a decrease in birth rate and in mortality and in morbidity, and an increase in life expectancy [3].

In Europe, the latest Eurostat data places the population over 65 as 19.7% of the total population. It is expected that by 2050, this will reach 30%. There is also significant feminization of old age, as there are double the number of women compared to men at 85 years of age.

According to the National Statistical Institute mortality tables, the life expectancy at birth of Spanish women is 87.7 years, and that of men is 80.4 years (the highest in Europe) [3,4].

Given the importance and social impact that this continuous increase of older people in Spain is having, this population group is being studied to better understand the ageing process and guarantee the health and quality of life of older people [5,6,7].

Ageing involves changing and acquiring knowledge and experiences that need adaptation and development at a personal and social level [1]. In this process, there are changes that worsen the state of health and physical fitness, causing a deterioration of organic functions such as physical, psychological and social functionality [1,8,9]. Ageing is not determined just by biological factors, but also by psychological, social and ecological factors, so whether this ageing is effective will depend on the ability of each person to adapt to the changes that occur, since this adaptation is a protective factor in the face of physical, mental and emotional decline [1,10].

The key is not just to age in any old way, but to do so in the best state of health by following a healthy lifestyle and participating in multiple activities [1,11]. Health is defined as the complete state of physical, mental and social well-being, not merely as the absence of disease—this is the fundamental basis for a population’s quality of life [12]. Healthy ageing looks beyond physical health; it tries to get older people motivated, to be satisfied with their life, to carry out physical activity and to have a relationship with their family and the environment [10].

Despite this, multiple health problems and diseases that affect older people are linked to a sedentary lifestyle, which is a worrying factor since a large proportion of older people are inactive. Inactivity is the fourth highest risk factor for mortality [13,14]. A sedentary lifestyle in old age increases the deterioration of muscular function [15] and the risk of suffering cardiovascular, metabolic, respiratory and degenerative diseases [7]. All this substantiates the great importance of reducing the sedentary nature of the lifestyle of older people, with the intention of reducing health problems, diseases, functional deterioration and dependency [7,16].

The intervention in ageing should be aimed at promoting personal growth and development by facilitating environments that prevent physical, psychological and social deterioration [17]. Therefore, it is essential to create a new mentality about ageing and to involve older people in society. The promotion of active ageing is necessary to produce health benefits through physical activity for this community [1].

Regarding physical activity, it is a very useful tool for achieving successful ageing, as well as for preventing adverse effects of ageing, such as the risk of mortality, many diseases, and functional and cognitive impairment. This is due to the fact that it intervenes in the bio-psycho-social factors of older people, with the purpose of ageing with greater functional autonomy and better integration into society [7,18,19].

In short, older people guided through active ageing through physical activity not only improve their physical condition, but also improve their quality of life as they get older, reinforcing their physical, psychological and social well-being [7,19].

### 1.1. Autonomy and Functional Capacity in Older People

The demographic aging of the Spanish population, according to WHO data and coinciding with global and European data, is the main cause of the increase in older people with functional dependency. Therefore, the government and public institutions have developed social, health and economic policies with the intention of preventing dependency and promoting the autonomy of older people. Specifically, an active ageing policy based on physical and leisure activity is being implemented, including relevant factors for achieving personal autonomy and reducing functional dependency in old age [19].

This decrease in functional independence in older people may be due to deterioration of muscle strength and mass, balance and cardiovascular endurance [15].

The assessment of functionality and degree of dependency allows older people to prepare specific plans for their self-care and to boost their motivation, with the aim of producing independence and autonomy in their daily lives [16]. For this reason, the health of older people should not be estimated based only on the presence or absence of disease, but rather according to functionality [20].

In relation to the activities of daily living, carrying them out is fundamental to the conservation of the physical, mental and social capacities of older people [21]. On the one hand, the Barthel Index (ADL) [22] focus on self-care and mobility, giving older people autonomy and independence to live without needing help from other people. On the other hand, Instrumental Activities of Daily Living Scale (IADL) [23] requires greater personal autonomy, and the ability to make decisions and solve problems that may arise in everyday life [24].

Consequently, regular physical activity should be promoted to preserve functionality and performance in the activities of daily living of older people [7]. Many investigations have shown the benefits of strength training to the autonomy of older people. Strength training is a very important factor in avoiding sarcopenia, optimizing functional capacity, and favoring the performance of the activities of daily living [25]. In addition, the improvement of lower body strength is of great importance for older people, since it provides them with balance and security when performing the ADL [26].

As for the physical exercise programs, those of the multi-component type where strength, resistance, balance and gait training are combined, are the most effective interventions to improve functional capacity and independence in performing the ADL in older people [27]. In addition, if these exercise programs are carried out systematically, there is an increase in strength, flexibility, aerobic capacity and balance in older people, contributing to the maintenance and improvement of their functional capacity [28].

### 1.2. Frailty and Risk of Falls in Older People

In the ageing process, frailty develops in older people, increasing the risk of adverse events such as functional impairment, dependency and falling [6,29]. Frailty in older people is considered a biological condition in which there is a poor response by several physiological systems to maintaining homeostasis after a stressful event [30]. As for falls, these can have physical and psychological consequences in older people, such as a fear of falling again, difficulties in walking, decreased functional capacity, needing help in performing the IADL, anxiety and depression [31].

Walking speed is a valid tool for predicting frailty in older people, which can show significant differences between healthy and frail older people since it is a variable that affects both falls and functional and cognitive deterioration [32,33]. Also, the timed up and go test is an important predictor of the risk of falls in older people [34].

Numerous investigations have proven the positive effects of physical activity on reducing adverse events caused by frailty in older people [29], since physical exercise produces improvements in strength, balance, autonomy and safety, in addition to reducing functional decline, illnesses, risk of falls, sarcopenia and disability [6,15,35]. The results of the study conducted by Mañas et al. [36] revealed that performing a physical activity of moderate to vigorous intensity is related to a greater decrease in frailty in older people when compared to the performance of a physical activity of low intensity or a sedentary lifestyle.

Multicomponent physical exercise programs are effective interventions for health and physical condition improvement in frail older people, delaying adverse events such as the risk of falls and functional impairment [15,27]. Along the same line, programs focused on strength and balance work are very effective in improving functional independence and preventing falls in older people [36]. In effect, training programs can lead to functional autonomy and the maintenance of strength and flexibility, key factors that contribute to the decrease in the risk of falls [33].

Therefore, the objectives of this study have been established based on this theoretical framework: (1) determine the differences in autonomy in the ADL and the IADL, functional capacity, frailty and risk of falls among the active group and the sedentary group in the sample, and (2) analyze the individual associations of functional capacity, frailty and risk of falls with autonomy in the ADL the IADL in the active group.

## 2. Methods

The research carried out was transversal and a descriptive and correlational design was used. The study variables were autonomy in the ADL, autonomy in the IADL, functional capacity, frailty and the risk of falls. The analysis variables were being in the experimental group (active) or the control group (sedentary).

### 2.1. Sample

The sampling was non-probabilistic but used for convenience. A total of 139 older people participated in this study, aged between 65 and 87 years (*M* = 73.1; standard deviation (*SD*) = 5.86). Of these, 100 were women (71.94%) and 39 were men (28.06%). The active group consisted of 69 older people (49.6%), and the sedentary group consistect of 70 (50.4%).

The sample came from four associations of older people based in social centers located in areas with a medium to low socio-economic level in the city of Huelva. The older people in the active group participated in a physical activity program launched by the collaboration between the City Council and the University of Huelva, while the older people in the sedentary group carried out other types of activities in these social centers. They have been classified as sedentary through the questionnaires applied in the comprehensive geriatric assessment notebook that includes the (International Physical Activity Questionnaire (IPAQ,) but this was not the main objective of this study.

In this work, the following criteria were established as inclusion criteria for delimitation of the population: older people between 65 and 90 years of age who perform some type of activity in the aforementioned social centers. If they attended the physical activity program, then they belonged to the experimental group (active), and if they performed any other type of activity, they were part of the control group (sedentary).

It should be noted that throughout the process of data collection and gathering, the current ethical and legal standards for research with human beings and for data protection were followed. The study was carried out in collaboration with the city of Huelva. They were responsible for compiling the mandatory documentation to participate in physical activity programs. The documents required were: medical certificate that enables the elderly to perform physical activity and informed consent with the research data.

### 2.2. Instruments

The instrument used to assess the autonomy in the ADL of the older people is the Barthel Index in Spanish [37], which was adapted from the original Barthel Index [21]. This scale assesses the ability to perform 10 basic activities of daily living on a dependent or independent basis: self-feeding, moving from a chair to a bed and back, personal hygiene and grooming, toilet hygiene, bathing and showering, transferring, going up and down stairs, dressing, and maintaining bowel and urinary control. Its score ranges between 0 (completely dependent) and 100 (completely independent) and the response categories range between two and four alternatives with five-point intervals depending on the time taken to complete it and the need for help in carrying it out. The lower the score, the greater the dependency, and the higher the score, the greater the independence. The psychometric properties of this scale show reliability and validity. For internal consistency, the Cronbach’s alpha coefficient was 0.965. For the interpretation of the Barthel Index, the results are grouped into the categories proposed by Shah, Vanclay and Cooper [38]: total dependency (0–20), severe dependency (21–60), moderate dependency (61–90), low dependency (91–99) and independence (100).

To assess the autonomy in the IADL of older people, the Spanish version of the Lawton IADL Scale [39] was used, which was adapted from the original Instrumental Activities of Daily Living Scale [22]. This scale consists of 8 items: the ability to use the telephone, to shop, to prepare food, to look after the home, to wash clothes, to use means of transport, to be responsible for medication, and to manage economic matters. The answers to each item can be 0 (dependent) or 1 (independent). The final score is the sum of the value of all the responses, ranging from 0 (maximum dependency) to 8 (total independence). The psychometric properties of this scale show reliability and validity. For internal consistency, the Cronbach’s alpha coefficient was 0.854.

The Vivifrail instrument [15] was used to evaluate the functional capacity, frailty and risk of falls of older people. The Vivifrail consists of the Short Physical Performance Battery Test (SPPB) and the Fall Risk Test: Timed Up and Go and Walking Speed Test (6 m). The SPPB test to determine the level of frailty is a composite of the following three separate measures: balance test (one foot next to the other, semi-tandem position and tandem position); walking speed test (4 m) and getting up from a chair. The psychometric properties of this scale show reliability and validity. For internal consistency, the Cronbach’s alpha coefficient was 0.767.

### 2.3. Process

First, the relevant permits were obtained from the City Council and the University of Huelva for the study to be carried out. Then, after being given the information sheet on the project, titled “Intergenerational programme of physical activity and improvement of health-related quality of life: university training and social transfer”, the older people of the Social Centers who wanted to participate had to sign an informed consent form.

Later, a battery of instruments was submitted for evaluation, where socio-demographic, clinical and psycho-social measures were collected. After completing these questionnaires, they were checked to make sure that they were properly filled in. Finally, in the following weeks, the Vivifrail instrument was passed to the study sample to assess their functional capacity, frailty and risk of falls.

### 2.4. Statistical Analysis

First, a descriptive analysis was performed to compare the study variables between the active group and the sedentary group from the sample. The Chi square test (*x*^2^) was used to analyze the homogeneity of the groups based on belonging to the active group or the sedentary group, and to confirm that the active group and the sedentary group were matched for gender, which was statistically insignificant (*p* = 0.006). Then, a multivariate analysis of variance (MANOVA) was performed to compare the mean scores of the active group and the sedentary group for the study variables. The MANOVA allows dependent variables to be correlated; it is better able to detect differences between groups than an ANOVA. Third, several analyses were performed to assess the association of the study variables in the active group. Pearson correlations were made from these variables, and also from gender to identify their roles as potential confounders, but since there was no correlation, it was not used. Then, to determine the relationship between the predictive factors of the Barthel Index and the Lawton and Brody Scale for the active group, linear regressions were performed. The level of significance was set at *p* < 0.05. The statistical package SPSS Statistics 21.0 (IBM, University of Chicago, USA) for Windows was used to build the database and perform the subsequent statistical analysis.

## 3. Results

Table 1 shows the descriptive results for the qualitative variables of the study according to whether they belong to the active group or the sedentary group. In all of them, statistically significant differences were found.

A MANOVA was applied to determine if there were differences in the study variables between the active group and the sedentary group. A significant effect was found after multivariate contrast was performed: *V* = 0.24, *F* (5, 137) = 8.58, *p* < 0.001. Table 2 shows that gender did not have a statistically significant effect: *x^2^* = 7.72, *p* = 0.006. In addition, older people belonging to the active group had better values for all the variables than those in the sedentary group, producing statistically significant differences for each of them, of practically zero size (*d* < 0.3). Specifically, in the Barthel Index: *F* (1, 137) = 12.96, *p* < 0.001; in the Lawton and Brody Scale: *F* (1, 137) = 14.94, *p* < 0.001; in the SPPB: *F* (1, 137) = 30.19, *p* < 0.001; in the 6-m Timed Walk: *F* (1, 137) = 24.15, *p* < 0.001; and in the Timed Up and Go Test: *F* (1, 137) = 25.54, *p* < 0.001.

Table 3 shows the correlations between gender values, the Barthel Index, the Lawton and Brody Scale, the SPPB, the 6-m Timed Walk and the Timed Up and Go Test for the active group. Gender is not correlated with the other variables. The Barthel Index correlates significantly with the Lawton and Brody Scale (*p* < 0.01), the SPPB (*p* < 0.01) and the Timed Up and Go Test (*p* < 0.01). While the Lawton and Brody Scale has a significant correlation with the SPPB (*p* < 0.01), the 6-m Timed Walk (*p* < 0.01) and the Timed Up and Go Test (*p* < 0.01). For its part, the SPPB correlates significantly with the 6-m Timed Walk (*p* < 0.01) and the Timed Up and Go Test (*p* < 0.01). Finally, the 6-m Timed Walk has a significant correlation with the Timed Up and Go Test (*p* < 0.01).

Table 4 shows the linear regressions between the Vivifrail, the Barthel Index and the Lawton and Brody Scale in the active group. This analysis allows the frailty and risk of falls variables measured in the Timed Up and Go Test to be identified as significant predictors of autonomy in the ADL, measured by the Barthel Index as 15% (Δ Adj. *R^2^* = 0.15). It was also found that the functional capacity variable measured in the SPPB is a significant predictor of autonomy in the IADL, measured by the Lawton and Brody Scale as 22% (Δ Adj. *R^2^* = 0.22).

## 4. Discussion

In this study, the influence of physical activity on improving functional capacity and autonomy in the ADL and the IADL has been shown, as well as its influence on reducing the frailty and risk of falls for the older people in the sample. When comparing the two intervention groups, statistically significant differences were found: older people in the active group have better values for all the variables evaluated compared to those in the sedentary group. This may be because physical activity produces an improvement in physical condition and functional capacity, as well as a lower risk of suffering from health problems and multiple diseases due to a sedentary lifestyle [7].

Another relevant finding is that, in the active group, it was found that functional capacity is a significant predictive variable of autonomy in the IADL by 22%, while frailty and the risk of falls are significant predictors of autonomy in the ADL by 15%. In addition, all variables have significant correlations with each other, except for the Barthel Index values and the values for the 6-m Timed Walk.

First, considering the functional capacity of this study’s sample, older people in the active group have a higher value than those in the sedentary group since regular physical activity is an effective tool for preserving the functional motor capacity of older people, thanks to having a healthy lifestyle [7].

Along the same lines, we find other studies, such as that carried out in [40], which show that the experimental group (who carried out a physical exercise program for 20 weeks) had an improvement in functional autonomy in the post-test compared to the pre-test values for the same group, and compared to the values for the control group, highlighting that strength training improves the functional autonomy of older people by encouraging the performance of everyday life activities. Similarly, in the study by Rodríguez-Berzal and Aguado [41], older people carried out a training program for eight weeks, with a frequency of two 25 min sessions per week, using exercises similar to the basic activities of daily living with one’s own body weight, resulting in an increase in functional capacity due to a significant improvement in lower body strength and balance. In contrast, in the study by Feijó et al. [42], no statistically significant differences were found between the active and sedentary groups of older people for the functional and walking tests, although this could be due to not having controlled for the intensity of the physical activity performed by the active group.

To recap the functional capacity, on the one hand, for this study’s sample’s active group, 75.4% had a minimal functional limitation, 21.7% had a low limitation, 2.9% had a moderate limitation and none had a serious limitation. On the other hand, for the sedentary group, 40% had a minimal functional limitation, 38.6% had a low limitation, 12.9% had a moderate limitation and 8.6% had a serious limitation. These data show that older people who do physical activity are less likely to have functional limitations, while those who are sedentary or inactive are more likely to suffer from this type of limitation. Similarly, in the study by Velasco et al. [8] (2015), more than 50% of older people in the sample had sufficient functional capacity (that is, total independence or low dependency). In contrast, 35% of older people had a significant functional deficit.

Secondly, with respect to carrying out activities of daily living, older people belonging to the active group showed greater autonomy than older people in the sedentary group. These data agree with those of the study carried out by De Dios and Martínez [43], where, after an intervention program based on walking, statistically significant differences were found in the performance of the ADL (*p* = 0.007) and by the improvement of the functional independence of the sample’s older people. Therefore, it can be assumed that carrying out physical activity promotes autonomy when performing these types of everyday activities [7,25], which are essential to maintaining the physical, mental and social capacities of older people [23]. Following the idea of Cerri [19], one should start working with older people with the assumption that the older person considered dependent has not lost their autonomy and that they need help to obtain it, since autonomy changes over time and with context through interactions with others.

In this study, a statistically significant correlation has been established between the Barthel Index and the Lawton and Brody Scale, as in the study carried out by Franco [10], since both instruments assess autonomy in the performance of the activities of daily living. Likewise, in the study by Marinês et al. [44], when linking both variables, it was found that among older people who performed the ADL independently, there was a significant percentage that required assistance to perform the IADL.

When analyzing the autonomy in the ADL for this study’s sample, on the one hand, in the active group, 94.2% were independent, 1.4% had a low dependency, 4.3% had a moderate dependency and nobody had a severe or total dependency. On the other hand, in the sedentary group, 71.4% were independent, 2.9% had a low dependency, 17.1% had a moderate dependency, 2.9% had a severe dependency and 5.7% were totally dependent. This greater proportion of autonomy in the ADL of the active group may be due to participation in the physical activity program, given that they are the most effective interventions to promote the performance of the ADL in older people [27].

Recent research has also analyzed the sample based on its autonomy in performing the ADL. In the study by Córdoba et al. [11], the results reveal that 40.9% of the older people in the sample were independent in all the ADL, 19.7% were independent in all but one or two, and that there was no older person with dependency in all. In the study by Franco [10], after the analysis of the Barthel Index, it was found that 30% of the sample had severe dependency, 40% had moderate dependency and the rest had low dependency in the performance of the ADL. However in the study by Quintero and Cerquera [20], high levels of independence were found in the performance of the ADL (specifically 62% of the sample of older people).

As for gender, it was not shown to be an influential variable for the functional capacity of the sample of this study, as there was a greater proportion of women than men. However, in many studies, gender has had an influence on the functional capacity of older people. In the study by Leirós-Rodríguez et al. [45], gender was a predictor of the functional limitations of the study sample, since men had a lower functional limitation than women for walking, climbing stairs, stooping and carrying a weight. On the other hand, in the study by Lozano et al. [46], for the most severe state of functional dependency, men had more limitations than women. Despite this, the women in the sample were more likely to go from a low to a moderate or severe functional dependency, while men were more likely to go from a low or severe functional dependency to no dependency. The results of the study conducted by Laguado et al. [16] show that, for the older people in the sample, low dependency predominates in males and independence predominates in females.

As for age, it also did not show a significant correlation with functional capacity, since it was not possible to establish age groups due to the differences in subjects that would have existed in each of these groups. However, the relationship between functional dependency increasing with age has been shown in a large fraction of research. In that by Leirós-Rodríguez et al. [45], it was shown that women had greater functional limitations from 75 years of age and men from 85 years of age. In Franco’s study [10], it was shown that the increase in functional dependency occurred after 80 years of age. This relationship was also significant in the study by Silva et al. [31], showing that functional independence is lost as age goes up.

Third, the results for frailty show that older people in the active group had lower frailty than those in the sedentary group. This implies that low physical activity is a factor related to frailty in older people [47], and that carrying out moderate to vigorous physical activity is a very effective resource for reducing the adverse effects that frailty causes [36] by producing an improvement in strength, balance, autonomy, and a reduction in sarcopenia and the risk of falls in older people [6]. Similar results were obtained in the study by De Dios and Martínez [43], where older people who performed the intervention focused on walking reduced their frailty, obtaining a significant improvement in balance, coordination, stability, agility and walking speed.

For falls, older people in the active group of this study have a lower risk than older people in the sedentary group. Therefore, it has been verified that interventions focused on performing physical exercise are linked to a lower risk of falls in older people [48,49]. Similarly, in the study by Vidarte et al. [28], older people who performed an exercise program based on strength work, flexibility and walking activities (experimental group) obtained statistically significant improvements in balance (*p* < 0.001) compared to the control group, which resulted in a reduction in the risk of falls.

In this study’s sample, older people in the active group had a 10.1% risk of falling; this rose to 38.6% in the sedentary group. In another study, the frequency of falls in the total sample was 35.6%, a significantly high value [34]. In the study by Silva et al. [31], the fraction of falls was 33.3%, and the majority of these were among women between 60 and 79 years old.

In this study, in the active group, a significant correlation was found between having a lower risk of falls and autonomy in the performance of both the IADL and the ADL. These data are consistent with the study by Kulzer-Homann et al. [34], in which older people who had difficulty performing at least one IADL were 78% more likely to fall than those who had no difficulty, while those who had difficulty in at least one ADL are 36% more likely to fall; that is, it was noted that not performing an ADL did not present a significant increase in the risk of falling, while not performing an IADL did significantly increase this risk.

In this case, age and gender were also not shown to be predictors of falls in this study, as in the study by Silva et al. [31], where no significant relationships were found between older people who had a fall and both analysis variables. In contrast, in the study by Kulzer-Homann et al. [34], it was shown that people 80 years old and older were 46% more likely to fall than those who were between 60 and 79 years old. In addition, women were 55% more likely to fall than men, although this could be because women have a longer life expectancy than men, although greater disability.

In short, it is absolutely necessary to achieve a culture which integrates active ageing, which must come with a change of mentality and attitude of the entire population to involve older people in all areas of society and to avoid their social exclusion [1]. It is also very important to end the sedentary lifestyle and promote the practice of systematic physical activity and healthy lifestyles in older people in order to improve their health and quality of life, as well as to prevent all kinds of adverse events and diseases that are produced [11,43,50,51].

In future studies, it would be advisable to use a quasi-experimental design in which measurements are made before (pre-test) and after (post-test) the intervention, in order to determine if the physical activity program is really the cause of the statistically significant differences found in the variables assessed in the experimental group (active) and the control group (sedentary). In addition, since an intentional non-probabilistic sampling was used, it was not possible to take into account either age or gender when analyzing the various variables, as there was a greater fraction of women than men, and different age groups could not be used. Therefore, in future research, the sample should be composed while taking into account both variables so that there is uniformity among them, and they could thus be used as analysis variables in the study. Also, a probabilistic sampling technique should be used and the sample size increased so that the results obtained could be extrapolated to the general population.

## 5. Conclusions

In this study, older people in the active group have better values for all the variables assessed compared to the sedentary group, showing statistically significant differences. Consequently, it has been possible to verify the influence of physical activity on improving functional capacity, increasing autonomy in performing the ADL and the IADL, and reducing frailty and risk of a fall in the older people in the sample. Further, in the active group, it was found that functional capacity is a significant predictive variable of autonomy in the IADL by 22%, while frailty and the risk of falls are significant predictors of autonomy in the ADL by 15%.

## Figures and Tables

**Table 1 ijerph-17-01006-t001:** Qualitative descriptive variable results of the study.

Instruments	Dimensions	Active, n (%)	Sedentary, n (%)
Barthel Index	Independence	65 (56.5%)	50 (43.5%)
	Low Dependence	1 (33.3%)	2 (66.7%)
	Moderate Dependence	3 (20%)	12 (80%)
	Severe Dependence	0	2 (100%)
	Total Dependence	0	4 (100%)
SPPB	Minimun Limitation	52 (65%)	28 (35%)
	Slight Limitation	15 (35.7%)	27 (64.3%)
	Moderate Limitation	2 (18.2%)	9 (81.8%)
	Severe Limitation	0	6 (100%)
6-m Timed Walk Test	Normal	47 (67.1%)	23 (32.9%)
	Fragility	15 (34.1%)	29 (65.9%)
	Mobility Problems and Falls	5 (29.4%)	12 (70.6%)
	Adverse Events and Falls	2 (25%)	6 (75%)
Timed Up and Go Test	Normal	58 (62.4%)	35 (37.6%)
	Fragility	11 (26.2%)	31 (73.8%)
	High Risk of Falls	0	4 (100%)

Barthel Index: *x^2^*(4) = 13.683; *p* = 0.008. SPPB: *x^2^*(3) = 21.077; *p* < 0.001. 6-m Timed Walk Test: *x^2^*(3) = 17.559; *p* = 0.001. Timed Up and Go Test: *x^2^*(2) = 19.206; *p* < 0.001. Risk of falls: *x^2^*(1) = 15.196; *p* < 0.001. SPPB: Short Physical Performance Battery Test.

**Table 2 ijerph-17-01006-t002:** Multivariate analysis of the variance of the study variables based on belonging to the active or sedentary group.

Variable	Active	Sedentary	*p*	Size of the Effect
Sex, n (%)				
Women	57 (82.6)	43 (61.4)	0.006	
Men	12 (17.4)	27 (38.6)		
Barthel Index	104.06 (3.67)	93.07 (25.08)	<0.001	0.098
Lawton and Brody Scale	7.77 (0.52)	6.87 (1.86)	<0.001	0.086
Vivifrail				
SPPB	10.41 (1.61)	8.26 (2.83)	<0.001	0.181
6-m Timed Walk Test	1.15 (0.26)	0.93 (0.26)	<0.001	0.15
Timed Up and Go Test	8.5 (1.96)	11.32 (4.2)	<0.001	0.157

Mean (standard deviation) of quantitative variables are presented as shown. *p*-values are based on MANOVA (quantitative variables) or *x^2^* (categorical variable). The effect size is based on Cohen’s *d*. SPPB: Short Physical Performance Battery Test.

**Table 3 ijerph-17-01006-t003:** Pearson’s correlations between sex, Barthel Index, Lawton and Brody Scale, SPPB, 6-m Timed Walk Test and Timed Up and Go test in the active group.

	A	B	C	D	E	F
Sex (A)	1	−0.119	−0.058	−0.017	−0.11	0.09
Barthel Index (B)		1	0.309 **	0.339 **	0.212	−0.378 **
Lawton and Brody Scale (C)			1	0.466 **	0.390 **	−0.414 **
SPPB (D)				1	0.678 **	−0.539 **
6-m Timed Walk Test (E)					1	−0.637 **
Timed Up and Go Test (F)						1

** The correlation is significant at the 0.01 level (bilateral).

**Table 4 ijerph-17-01006-t004:** Linear regression between the predictive factors of the Barthel Index and Lawton and Brody Scale in the active group.

	*b*	*DT*	*β*	*p*
Barthel Index				
SPPB	0.66	0.35	0.29	0.06
6-m Timed Walk Test	−3.02	2.39	−0.21	0.21
Timed Up and Go Test	−0.67	0.28	−0.36	0.02
Lawton and Brody Scale				
SPPB	0.11	0.05	0.33	0.03
6-m Timed Walk Test	0.05	0.33	0.03	0.87
Timed Up and Go Test	−0.06	0.04	−0.22	0.13

*F* (3) = 5.04; *p* = 0.003. *F* (3) = 7.41; *p* < 0.001. SPPB: Short Physical Performance Battery Test.

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
