# Peer review of "Functional Ability, Frailty and Risk of Falls in the Elderly: Relations with Autonomy in Daily Living"

_ijerph, 2020, doi:10.3390/ijerph17031006_

Round 1

Reviewer 1 Report

Please present European statistic for the increase in elderly people - not Spanish.

The Introduction is too long - a reduction is needed...

Explain the "randomisation", why were the patients not randomised.

Describe the etic considerations more detailed, eg. was consent collected??

More details about the process are needed. How and when were the patients tested and by WHO? Were the testers blinded? etc....

The results are predictable according to how the patients are divided into the two groups.

The main problem in this study is that the patients are not randomised to the two groups.

Author Response

Good morning, we have made the requested corrections.
Please look at the attached document.

Thank you.

Reviewer 2 Report

The study includes some interesting concepts but, from my point of view, it should be reformulated starting from the level of physical activity the participants should achieve to be considered “active” or “sedentary”. Moreover, it is recommended to follow Consort guidelines  for the quality control of the paper. English should be revised by an expert.

Title

The title does not reflects what the authors have made in this research. It should be rewritten in order to make it fits with the two objectives.

Introduction

It is more common to use ADL as an abreviation of “basic activities of daily living”. Please, revise it. The introduction is too long and there are paragraphs, such as: “For the assessment of frailty and the risk of falling for older people, Izquierdo et al. [15] 129 recommend using the 6-metre timed walk and the Timed up and go tes…”, that should not be included in the introduction but in the method section. Please, rewrite it in order to be mor concise regarding the study objective and variables.

Method

Which were the other kind of activities carried out by the sedentary group? Please, specify. It is not specified if the sample signed a written consent before enrolling in the study. Please, avoid spanish description of the outcome instruments. There is a lack of information about the characteristics of the active group in relation to their level of activity, the attendance to exercise sessions, the kind of exercise they were involved in. This is of relevance because its incidence in their autonomy. How do you clasify  a subject as “active”? Maybe its level of attendance to the eercise sessions was of one day/month? Moreover, the subjects from tha “sedentary group” were not engaged in any exercise group but maybe they made daily physical activities such as walking. The level of the physical activity should had been measured in order to distribute the sample in the two groups, for example, with the PASE scale or with accelerometry. This information should be controlled and included in the study. Please, avoid spanish text in the tables. According to your objectives, you should be able to resolve this answer: how active should I be to consider myself functional enough?

Results

Why correlations are made only in the active group? Correlations should be made only with cuantitative variables, so sex should not be included in the analysis.

Author Response

(The authors gave the same response as above.)

Reviewer 3 Report

First of all, I would like to congratulate the authors on this contribution. It is a simple, direct manuscript that can be helpful to the scientific community and the design of public health policies. The approach (lines 96-97) that health should be estimated by the functionality of the person, not only for lack of disease, is very interesting.

I suggest that the authors address these recommendations:

There is a previous conceptual question: what do the authors consider inactivity? It would be interesting to better explain when an elderly person is considered sedentary or inactive. This could be explained in a sub-section, in the methodology, dedicated to variables.

The characteristics of the scales are listed below. This part could well be included in the new sub-section where the variables are better explained. Lines 65-66: The authors could indicate the percentage of the elderly population are inactive. Line 89: Who promotes this policy? Does it apply in the same way throughout Spain? Lines 156-157: It would be good to concentrate the explanation of the variables and the nature of the variables in a new section within methodology. Lines 171-172: The control group also includes those that do another type of activity. Therefore, there are people who are not really sedentary. How have the authors controlled this? How is this controlled in results, differences or non-existence in certain scale areas? In the literature, some authors advise using the Katz Index, an international scale most commonly used to assess the functional situation of the elderly (regardless of their pathology). And the Barthel Index, to evaluate functionality in elderly people with cerebrovascular pathologies... why did the authors make this option? This should be better explained. Line 276: It's necessary to translate the note in the Table into English. As the authors say, one of the most important limitations is not to analyze the sample by age. This should be better explained in the new Subsection of Variables in Methodology.

Author Response

(The authors gave the same response as above.)

Round 2

Reviewer 2 Report

I accept all the changes made by the authors.